**Data Availability Statement:** All relevant data are within the manuscript and its Supporting information files.

# Prevalence and predictors of death and severe disease in patients hospitalized due to COVID-19: A comprehensive systematic review and meta-analysis of 77 studies and 38,000 patients

**Kunchok Dorjee**[1]*, **Hyunju Kim**[2], **Elizabeth Bonomo**[1], **Rinchen Dolma**[3]

**1** School of Medicine Division of Infectious Diseases, Center for TB Research, Johns Hopkins University, Baltimore, Maryland, United States of America, **2** Department of Epidemiology, Bloomberg School of Public Health, Johns Hopkins University, Baltimore, Maryland, United States of America, **3** Center for Alcohol and Addiction Studies, Brown University School of Public Health, Brown University, Providence, Rhode Island, United States of America

* kdorjee1@jhmi.edu

## Abstract

### Introduction

Progression of COVID-19 to severe disease and death is insufficiently understood.

### Objective

Summarize the prevalence of risk factors and adverse outcomes and determine their associations in COVID-19 patients who were hospitalized.

### Methods

We searched Medline, Embase and Web of Science for case-series and observational studies of hospitalized COVID-19 patients through August 31, 2020. Data were analyzed by fixed-effects meta-analysis using Shore's adjusted confidence intervals to address heterogeneity.

### Results

Seventy-seven studies comprising 38906 hospitalized patients met inclusion criteria; 21468 from the US-Europe and 9740 from China. Overall prevalence of death [% (95% CI)] from COVID-19 was 20% (18–23%); 23% (19–27%) in the US and Europe and 11% (7–16%) for China. Of those that died, 85% were aged≥60 years, 66% were males, and 66%, 44%, 39%, 37%, and 27% had hypertension, smoking history, diabetes, heart disease, and chronic kidney disease (CKD), respectively. The case fatality risk [%(95% CI)] were 52% (46–60) for heart disease, 51% (43–59) for COPD, 48% (37–63) for chronic kidney disease (CKD), 39% for chronic liver disease (CLD), 28% (23–36%) for hypertension, and 24% (17–33%) for diabetes. Summary relative risk (sRR) of death were higher for age≥60 years

**Funding:** Dr. Dorjee is supported by grants from the US National Institute of Allergy and Infectious Diseases of the National Institute of Health (Grant # K01AI148583); Johns Hopkins Alliance for a Healthier World (Grant # 80045453); STOP TB PARTNERSHIP TB REACH (Grant # 134126); the Pittsfield Anti-TB Association and dedicated private philanthropists.

**Competing interests:** The authors have declared that no competing interests exist.

[sRR = 3.6; 95% CI: 3.0–4.4], males [1.3; 1.2–1.4], smoking history [1.3; 1.1–1.6], COPD [1.7; 1.4–2.0], hypertension [1.8; 1.6–2.0], diabetes [1.5; 1.4–1.7], heart disease [2.1; 1.8–2.4], CKD [2.5; 2.1–3.0]. The prevalence of hypertension (55%), diabetes (33%), smoking history (23%) and heart disease (17%) among the COVID-19 hospitalized patients in the US were substantially higher than that of the general US population, suggesting increased susceptibility to infection or disease progression for the individuals with comorbidities.

## Conclusions

Public health screening for COVID-19 can be prioritized based on risk-groups. Appropriately addressing the modifiable risk factors such as smoking, hypertension, and diabetes could reduce morbidity and mortality due to COVID-19; public messaging can be accordingly adapted.

## Introduction

Coronavirus disease-19 (COVID-19) caused by severe acute respiratory syndrome- coronavirus-2 (SARS-CoV-2) that first emerged in Wuhan, China in late December 2019 has spread with such rapidity and efficiency that in less than 10 months, it has caused more than 36 million cases and million deaths globally [1]. Driven by an urgency to solve the crisis, studies are being published at an unprecedented pace. However, across the publications, prevalence of death, severe disease and their association with epidemiological risk factors have greatly varied [2, 3], with studies showing conflicting results for association of key risk factors such as sex [4–8], smoking [9–12], hypertension [4, 7, 8, 13, 14] and diabetes [4, 7, 8, 13, 14] with COVID-19 disease severity and death. Whether or how cardiovascular risk factors, especially prior hypertension, diabetes and heart disease are associated with acquisition of SARS-CoV-2 and progression to severe disease or death is not understood well [15–18]. Meta-analyses conducted so far on prevalence of epidemiological risk factors and association with disease progression were mostly based on studies from China [9, 11, 18–20] and many of the analyses on prevalence estimates included studies focused on critically ill patients [9, 19], which can overestimate the prevalence and affect generalizability of results. To our knowledge, none of the analyses were restricted to hospitalized COVID-19 patients. Restricting our analysis to hospitalized patients provides an efficient sampling frame to investigate disease progression in relation to risk factors.

Therefore, we undertook a comprehensive systematic review and meta-analysis to investigate the association between key epidemiological factors–age, gender, smoking, hypertension, diabetes, heart disease, chronic obstructive pulmonary disease (COPD), chronic kidney disease (CKD) and chronic liver disease (CLD)–and progression to death and severe disease in patients hospitalized due to COVID-19. We additionally compared the 1) the prevalence of risk factors and death in the US-Europe with that of China; 2) the prevalence of co-morbidities at baseline with the general population prevalence, and 3) prevalence of cardiovascular disease, COPD and CKD at baseline with corresponding organ injuries (acute cardiac injury, acute lung injury, and acute kidney injury) during hospital admission.

## Methods

### Literature search, study selection and data abstraction

We searched Medline, Embase, Web of Science and the WHO COVID-19 database to identify studies published through August 31, 2020 that investigated the risk of severe disease or death

in hospitalized patients with confirmed COVID-19 disease. We used search terms, 'coronavirus disease 19', 'COVID-19', 'severe acute respiratory syndrome coronavirus 2' and 'SARS-CoV-2' for COVID-19 and the string ((characteristics) OR (risk factors) OR (epidemiology) OR (prevalence) OR (intensive care) OR (ventilator) OR (mechanical ventilator) OR (mortality) OR (survivor*) OR (smoking) OR (smoker*)) AND ((COVID-19) OR (COVID) OR (coronavirus)) for studies published between December 15, 2019 and August 31, 2020. We started the search on March 18, 2020 with biweekly search thereafter and final search on August 31, 2020. We included case series and observational studies that described the prevalence of death or severe disease in adult population stratified by risk factors: age, sex, hypertension, diabetes, heart disease, COPD, CKD and CLD. We excluded studies that included non-consecutive patients or exclusively focused on pregnant women, children, and elderly patients. We excluded studies that exclusively studied critically ill patients from calculation of prevalence of death but included them for calculating the association of risk factors with death. Screening of abstracts and full-text reviews were conducted using Covidence (Melbourne, Australia).

## Risk factors and outcomes

Primary outcomes were prevalence of death and association of risk factors with death. We extracted data on death as recorded in the publications. We measured prevalence of severe disease and association with risk factors as secondary outcomes. We defined outcome as severe disease for any of 1) the study classified COVID-19 disease as severe or critical, 2) intensive care unit (ICU) admission, 3) acute respiratory distress syndrome, or 4) mechanical ventilation. Severe disease was defined by studies as respiratory rate$\geq$30 per minute, oxygen saturation$\leq$93%, and $PaO_2/FiO_2 < 300$ and/or lung infiltrates>50% within 24–48 hours [3]. Critical illness was defined as respiratory failure, shock and/or multiple organ dysfunction or failure [3]. Heart disease as a pre-existing condition was broadly defined by most studies as 'cardiovascular disease' (CVD). Additional outcomes were acute cardiac and kidney injury in the hospitalized patients that were defined as such by the studies.

## Statistical analysis

We calculated and reported summary estimates from fixed-effects models [21]. We assessed heterogeneity across studies using Cochran's Q-test ($\chi^2$ p value <0.10) [22] and $I^2$ statistics ($I^2 > 30\%$) [23]. In the presence of heterogeneity, we adjusted the confidence intervals for between-study heterogeneity using the method described by Shore et al. [24]. We presented the results from random effects meta-analysis as well. The meta-analysis was performed in Microsoft® Excel 2020 (Microsoft Corporation, Redmond, WA). We analyzed publication bias using funnel plots and Egger's tests. Quality of each study was assessed using the Newcastle-Ottawa assessment scales using the PRISMA guidelines. We calculated the following as a part of our analyses: 1) prevalence of severe disease or death, 2) prevalence of risk factors, and 3) relative risk for the association of age, sex, and comorbidities with outcome. When not reported or when unadjusted odds ratio was presented, we calculated the relative risk (95% CI) using the frequencies provided. Adjusted estimates were used where available. Case fatality risk (and case severity risk) for a specific risk factor was calculated as number of deaths (or severe disease) in patients with a risk factor out of all patients possessing the risk factor.

# Results

## Study characteristics

Initial search yielded 30133 citations. Articles were then screened (Fig 1). We identified 410 articles for full text review, of which 77 studies met inclusion criteria (Table 1) [4–8, 13, 14, 25–94]. The studies were conducted in: China (n = 35), USA (n = 18), Europe (n = 10), rest of Asia (n = 5) and Africa (n = 1). Two studies were prospective, five cross-sectional, and remaining retrospective in nature.

## Population and demographics

There were 38,906 total COVID-19 hospitalized patients including 21468 patients from the US and Europe (87% from the US), and 9740 patients from China. Median age was 59 years [IQR: 57–62 years; $I^2$ = 58%; n = 62 studies] and 48% [95% CI: 44–53%; $I^2$ = 98%; n = 41] were aged≥60. Fifty-nine percent [95% CI: 57–60%; $I^2$ = 98%; n = 75] of the patients were males.

## Prevalence of death and severe disease

We calculated an overall prevalence of death of 20% [95% CI: 18–23%; $I^2$ = 96%; n = 60], ranging from 1% to 38% across the studies, and of severe disease of 28% [95% CI: 24–33%; $I^2$ = 98%; n = 60] for all patients hospitalized due to COVID-19 (Tables 2 and 3). Data on

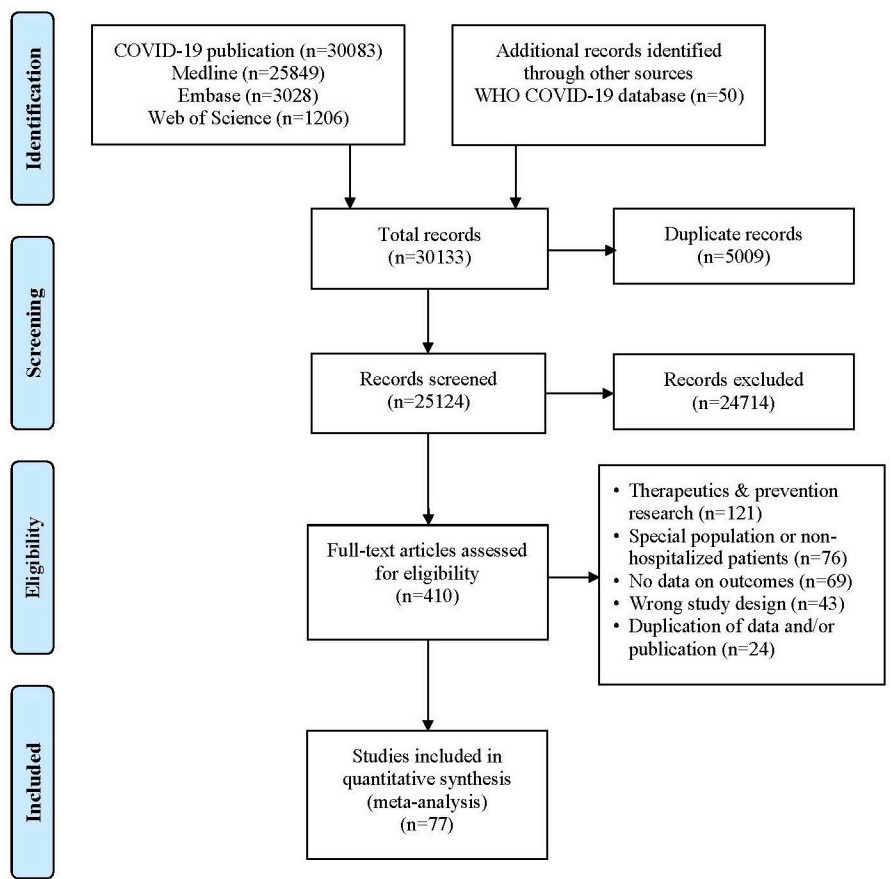

**Fig 1. PRISMA flow diagram for selection of studies.**

**Table 1. Characteristics of studies to determine prevalence of risk factors and death or severe disease and their associations in patients hospitalized for COVID-19 globally.**

| Author, year of publication (journal) | Country | Region | Study Period | Study Design | Size | Epidemiological Risk Factor | Outcome | Measures of Association |
|---|---|---|---|---|---|---|---|---|
| Aggarwal S et al., 2020 (Diagnosis) | USA | Des Moines | 3-1-2020 to 4-4-2020 | Retrospective | 16 | Age, sex, smoking, substance use, obesity, HTN, DM, CVD, COPD, CKD, Cancer | Prevalence of death and primary end point (death, shock, or ICU admission). Association of risk factors with outcome | Unadjusted RR calculated |
| Argenziano M. G et al., 2020 (BMJ) | USA | New York City | 3-11-2020 to 4-6-2020 | Retrospective | 1,000 | Age, sex, ethnicity, obesity, smoking, HTN, DM, CVD, COPD, CKD, cancer, HIV, viral hepatitis, cirrhosis | Association of risk factors with disease severity and death. | Adjusted HR |
| Brill S. E et al., 2020 (BMC Medicine) | UK | Barnet | 3-10-2020 to 4-8-2020 | Retrospective | 450 | Age, race, sex, smoking, HTN, DM, CVD, immunosuppression | Prevalence of death. | Unadjusted RR calculated |
| | | | | | | | Association of comorbidities with disease severity. | |
| Cao Z et al., 2020 (PLOS ONE) | China | Beijing | 1-21-2020 to 2-12-2020 | Retrospective | 80 | Sex, age, HTN, CVD, DM, COPD, smoking | Association of risk factors with disease severity. | Unadjusted RR calculated |
| CDC (MMWR) | USA | National | 2-12-2020 to 3-28-2020 | Retrospective | 5285 | Age, Current Smoker, DM, CVD, COPD, CKD, CLD | Prevalence of ICU admission. Association of risk factors with severe disease (ICU admission). | Unadjusted RR calculated |
| Chen G et al., 2020 (Journal of Clinical Investigation) | China | Wuhan | December 2019 to 01-27-2020 | Retrospective | 21 | Age, sex, Huanan sea food market exposure, HTN, DM | Prevalence of severe disease. Compared moderate and severe cases based on risk factors. | Unadjusted RR calculated |
| Chen J et al., 2020 (Journal of Infection) | China | Shanghai | 1-20-2020 to 2-6-2020 | Retrospective | 249 | Age, sex | Prevalence of ICU admission. Association of age and sex with ICU admission. | Adjusted OR reported for age and sex |
| Chen Q et al., 2020 (Infection) | China | Zhejiang province | 1-1-2020 to 3-11-2020 | Retrospective | 145 | Age, sex, smoking, exposure history, BMI, HTN, DM, COPD, CKD, Solid tumor, Heart disease, HIV infection | Prevalence of severe disease. Association of risk factors with severe disease. | Unadjusted RR calculated |
| Chen T et al., 2020 (BMJ) | China | Wuhan, Hubei | 1-13-2020 to 2-28-2020 | Retrospective | 274 | Age, sex, sea food market exposure, contact history, smoking HTN, DM, CVD, CHF, heart failure, cancer, HBV, HIV, CKD | Association of risk factors with death. | Unadjusted RR calculated |
| | | | | | | | Compared death and recovered group. Presently hospitalized patients excluded from study. | |
| Chilimuri S et al., 2020 (West J Emerg Med) | USA | New York City | 3-9-2020 to 4-9-2020 | Retrospective | 375 | Age, sex, ethnicity, HTN, DM, CVD, COPD, CKD, HIV/AIDS, CLD | Association of risk factors with disease severity and death. | Adjusted OR |
| | | | | | | | | reported for age, sex and comorbidities |
| Ciceri F et al., 2020 (Clinical Immunology) | Italy | Milan | 2-25-2020 to 5-1-2020 | Retrospective | 410 | Age, sex, ethnicity, BMI, HTN, CVD, DM, COPD, CKD, cancer | Prevalence of death. | Adjusted HR |
| | | | | | | | Association of risk factors with disease severity. | |

(*Continued*)

**Table 1.** (Continued)

| Author, year of publication (journal) | Country | Region | Study Period | Study Design | Size | Epidemiological Risk Factor | Outcome | Measures of Association |
|---|---|---|---|---|---|---|---|---|
| Cummings MJ et al., 2020 (The Lancet) | USA | New York City | 3-2-2020 to 4-1-2020 | Prospective | 257 | Age, sex, race, BMI, HTN, DM, chronic cardiac disease (CHD and CHF), CKD, smoking history, COPD, cancer, HIV, cirrhosis | Association of risk factors with death. | Adjusted HR |
| Deng Y et al., 2020 (Chin Med J) | China | Wuhan | 1-1-2020 to 2-21-2020 | Retrospective | 116 out of 964 | Age, sex, HTN, DM, Heart Disease, Cancer | Association of risk factors with death. | Unadjusted RR calculated |
| | | | | | | | Compared death and recovered group. | |
| | | | | | | | Presently hospitalized patients excluded from study. | |
| Du R-H et al., 2020 (ERJ) | China | Wuhan, Hubei | 1-25-2020 to 2-7-2020 | Retrospective | 179 | Age, sex, HTN, DM, CVD, TB, cancer, CKD or CLD | Prevalence of death. Association of risk factors with death. | Adjusted OR for age≥65 and CVD. Unadjusted RR calculated for other variables |
| Escalera-Antezana et al., 2020(Infez Med) | Bolivia | Nationwide | 3-2-2020 to 3-29-2020 | Retrospective | 107 | Age, HTN, CVD, DM, obesity, sex | Prevalence of death. | Adjusted OR |
| | | | | | | | Association of risk factors with disease severity. | reported for age, sex and risk factors |
| Feng Y et al., 2020 (AJRCCM) | China | Wuhan, Shanghai, Anhui province | 1-1-2020 to 2-15-2020 | Retrospective | 476 | Age, age groups, sex, Wuhan exposure, smoking, alcohol, HTN, anti-hypertensives, CVD, DM, cancer, COPD, CKD | Prevalence of death. Association of risk factors with severe disease. | Adjusted HR for HTN, CVD, DM. Unadjusted RR calculated for other variables |
| Ferguson J et al., 2020 (EID) | USA | Northern California | 03-13-2020 to 04-11-2020 | Retrospective | 72 | Sex, race, smoking, HTN, DM, CKD, Heart Disease, COPD | Prevalence of ICU admission. Association of risk factors with severe disease (ICU admission). | Unadjusted RR calculated |
| Galloway J.B et al., 2020 (Journal of Infection) | UK | London | 3-1-2020 to 4-17-2020 | Retrospective | 1,157 | Age, sex, ethnicity, cancer, CKD, DM, HTN, CVD, COPD | Prevalence of death. | Adjusted HR reported for age and sex |
| | | | | | | | Association of risk factors with disease severity. | |
| Garibaldi B et al., 2020 (Ann Intern Med) | USA | Maryland | 3-4-2020 to 6-27-2020 | Retrospective | 832 | Age, sex, alcohol, smoking, BMI, cancer, CVD, COPD, HTN, liver disease, CKD, HIV/AIDS DM | Association of risk factors with disease severity. | Adjusted HR |
| | | Washington DC | | | | | | reported for age, ethnicity and BMI |
| Giacomelli A et al., 2020 (Pharmacol Res) | Italy | Milan | 2-21-2020 to 4-20-2020 | Prospective | 233 | Sex, age, smoking, obesity | Prevalence of death. | Adjusted HR |
| | | | | | | | Association of risk factors with disease severity. | reported for sex, age, and obesity |
| Gold J et al, 2020 (MMWR) | USA | Georgia | 3-1-2020 to 3-30-2020 | Retrospective | 305 | Age, sex, race, HTN, DM, Heart Disease, COPD, CKD, Cancer | Prevalence of patient characteristics, death, and ICU. | Unadjusted RR calculated |
| Goyal P et al. 2020 (NEJM) | USA | New York City | 3-3-2020 to 3-27-2020 | Retrospective | 393 | Age, sex, race, smoking, HTN, DM, COPD, Heart Disease, Asthma | Prevalence of severe disease (mechanical ventilation). Association of risk factors with severe disease. | Unadjusted RR calculated |

(Continued)

**Table 1.** (Continued)

| Author, year of publication (journal) | Country | Region | Study Period | Study Design | Size | Epidemiological Risk Factor | Outcome | Measures of Association |
|---|---|---|---|---|---|---|---|---|
| Gregoriano C et al.,2020 (Swiss Medical Weekly) | Switzerland | Aarau | 2-26-2020 to 4-30-2020 | Retrospective | 99 | Age, sex, smoking, HTN, cancer, CVD, COPD, obesity, DM, rheumatic disease, organ transplant recipient, obstructive sleep apnea | Prevalence of disease endpoints (transfer to ICU and in-hospital mortalities). | Unadjusted OR |
| | | | | | | | Association of comorbidities with disease endpoints. | |
| Guan et al., 2020 (NEJM) | China | Nationwide | 12-11-2019 to 01-31-2020 | Retrospective | 1099 | Age, sex, smoking, exposure to transmission source, HTN, DM, CHD, CKD, COPD, Cancer, HBV, cerebrovascular disease, immunodeficiency | Prevalence of death, composite outcome, ((Death/MV/ICU) and severe disease. Association with severe disease and composite outcome. | Unadjusted RR calculated |
| Guan Wei-Jie, 2020 (ERJ) | China | Nationwide | 12-11-2019 to 1-31-2020 | Retrospective | 1590 | Age, sex, smoking, CKD, COPD, HTN, DM, CVD, Cancer, HBV | Prevalence of patient characteristics, death and composite outcome (Death, ICU, MV). | Adjusted HR |
| Hewitt J et al., 2020 (Lancet) | UK | Nationwide (UK), | 2-27-2020 to 4-28-2020 | Prospective | 1,564 | Age, sex, smoking, DM, HTN, CVD, CKD | Prevalence of death. | Adjusted HR |
| | Italy | Modena (Italy) | | | | | Association of risk factors with disease severity. | |
| Hsu H. E et al., 2020 (Morbidity and Mortality Weekly Report) | USA | Boston | 3-1-2020 to 5-18-2020 | Retrospective | 2,729 | Age, sex, ethnicity, COPD, cancer, CKD, cirrhosis, CVD, DM, HIV/AIDS, HTN, obesity, substance use disorder | Association of risk factors with disease severity. | Unadjusted RR calculated |
| Hu L et al., 2020 (CID) | China | Wuhan | 1-8-2020 to 2-20-2020 | Retrospective | 323 | Age, sex, current smoker, HTN, DM, CVD, COPD, CKD, CLD, Cancer | Prevalence of severe (severe and critical) disease. Association of risk factors with disease severity. | Unadjusted RR calculated |
| Huang C et al., 2020 (The Lancet) | China | Wuhan | 12-16-2020 to 1-2-2020 | Prospective | 41 | Age, sex, Huanan seafood market exposure, smoking, HTN, DM, CKD, COPD, CVD, Cancer, CLD | Association of risk factors with severe disease (ICU care). | Unadjusted RR calculated |
| Hur K et al., 2020 (Otolaryngol Head Neck Surg) | USA | Chicago | 3-1-2020 to 4-8-2020 | Retrospective | 486 | Age, sex, BMI, smoking, HTN, DM, CVD, COPD, cancer, immunosuppression, CKD, | Association of risk factors with disease severity. | Adjusted HR (for age, sex, ethnicity BMI, HTN, smoking) |
| Iaccarino G et al., 2020 (Hypertension) | Italy | Nationwide | 3-9-2020 to 4-9-2020 | Cross-sectional | 1,591 | Age, sex, HTN, obesity, DM, COPD, CKD, CVD, cancer | Prevalence of death. | Adjusted OR |
| | | | | | | | Association of risk factors with disease severity. | |
| Inciardi R et el., 2020 (Eur Heart J) | Italy | Lombardy | 3-4-2020 to 3-25-2020 | Retrospective | 99 | Sex, smoking, HTN, DM, coronary artery disease, COPD, CKD, cancer | Prevalence of death. Association of risk factors with death. | Unadjusted RR calculated |
| Jang J.G et al., 2020 (Journal of Korean Medical Science) | South Korea | Daegu | 2-19-2020 to 4-15-2020 | Retrospective | 110 | Age, sex, CVD, cerebrovascular disease, COPD, dementia, DM, HTN, connective tissue disease liver disease, malignancy, Parkinson's disease | Association of risk factors with disease severity and death. | Adjusted OR |

*(Continued)*

**Table 1.** (*Continued*)

| Author, year of publication (journal) | Country | Region | Study Period | Study Design | Size | Epidemiological Risk Factor | Outcome | Measures of Association |
|---|---|---|---|---|---|---|---|---|
| Javanian M et al., 2020 (Rom J Intern Med) | Iran | Mazandaran province | 2-25-2020 to 3-12-2020 | Retrospective | 100 | Age, sex, HTN, DM, CVD, CKD, cancer, CLD | Prevalence of death. Association of risk factors with death. | Unadjusted RR calculated |
| Kalligeros M et al., 2020 (Obesity Journal) | USA | Rhode Island | 2-17-2020 to 4-5-2020 | Retrospective | 103 | Age, sex, ethnicity, smoking, BMI (obesity), cancer, CKD, cirrhosis, DM, heart disease (CVD), HTN, lung disease (COPD), transplant | Association of risk factors with disease severity. | Adjusted OR (for age, sex, ethnicity, BMI, DM, HTN, heart disease, lung disease) |
| Khalil K et al., 2020 (Journal of Infection) | UK | London | 3-7-2020 to 4-7-2020 | Prospective | 220 | Age, sex, ethnicity, smoking, COPD, CVD, HTN, hyperlipidemia, DM, CKD, CVA, dementia, liver disease, cancer | Prevalence of death. Association of risk factors with disease severity. | Unadjusted RR calculated |
| Khamis F et al., 2020 (Journal of Infection and Public Health) | Oman | Muscat | 2-24- 2020 to 4-24-2020 | Retrospective | 63 | Age, sex, smoking, substance use, HTN, DM, CKD, CVD | Prevalence of severe disease and death. Association of risk factors with disease severity. | Unadjusted RR calculated |
| Lendorf M.E et al., 2020 (Danish Medical Journal) | Denmark | North Zealand | 3-1-2020 to 5-18-2020 | Retrospective | 111 | Age, sex, BMI, cancer, HTN, CVD, COPD, immunosuppression, CKD, DM, smoking | Association of risk factors with disease severity and death. | Unadjusted RR calculated |
| Li X et al., 2020 (J Allergy Clin Immunol) | China (Wuhan, Hubei) | Wuhan, Hubei | 1-26-2020 to 2-5-2020 | Retrospective | 548 | Age, sex, smoking, HTN, DM, Heart Disease, CKD, Cancer, COPD | Prevalence of death and severe disease. Association of risk factors with severe disease. | Unadjusted RR calculated |
| Liu S et al., 2020 (BMC Infectious Diseases) | China | Jiangsu Province | 1-10-2020 to 3-15-2020 | Retrospective | 625 | Sex, age, HTN, DM, CVD | Association of risk factors with disease severity. | Adjusted OR (for age and HTN) |
| Liu W et al. 2020 (Chin Med J) | China | Wuhan | 12-30-2019 to 01-15-2020 | Retrospective | 78 | Age, sex, smoking history, exposure to Huanan seafood market, HTN, diabetes, COPD, cancer | Compared progression group and stabilization group. Progression group defined by progression to severe or critical disease or death. | Unadjusted RR calculated |
| Nikpouraghdam M et al., 2020 (J Clin Virol) | Iran | Tehran | 2-19-2020 to 4-15-2020 | Retrospective | 2,964 | Age, sex, DM, COPD, HTN, CVD, CKD, cancer | Prevalence of death. Association of risk factors with disease severity. | Adjusted OR |
| Nowak B et al., 2020 (Pol Arch Intern Med) | Poland | Warsaw | 3-16-2020 to 4-7-2020 | Retrospective | 169 | Sex, smoking, HTN, DM, CVD, COPD, CKD, AKI, cancer | Prevalence of death. Association of risk factors with death. | Unadjusted RR calculated |
| Okoh A.K et al., 2020 (Int J Equity Health) | USA | Newark | 3-10-2020 to 4-20-2020 | Retrospective | 251 | Age, sex, ethnicity, BMI, HTN, DM, CVD, COPD, HIV, CKD, cancer | Prevalence of death. Association of risk factors with disease severity and death. | Adjusted OR |
| Palaiodimos L et al., 2020 (Metabolism) | USA | New York | 3-9-2020 to 3-22-2020 | Retrospective | 200 | Age, sex, race, smoking, HTN, DM, coronary artery disease, COPD, CKD, cancer | Prevalence of death. Association of risk factors with death. | Adjusted OR (provided by the study) |

(*Continued*)

**Table 1.** (Continued)

| Author, year of publication (journal) | Country | Region | Study Period | Study Design | Size | Epidemiological Risk Factor | Outcome | Measures of Association |
|---|---|---|---|---|---|---|---|---|
| Pellaud C et al., 2020 (Swiss Medical Weekly) | Switzerland | Fribourg | 3-1-2020 to 5-10-2020 | Retrospective | 196 | Sex, age, HTN, DM, obesity, CVD, COPD, cancer, immunosuppression, smoking | Prevalence of death. | Unadjusted RR calculated |
| | | | | | | | Association of risk factors with disease severity. | |
| Richardson S et al., 2020 (JAMA) | USA | New York | 3-1-2020 to 4-4-2020 | Retrospective | 5700 | Age, sex, race, smoking, HTN, DM, COPD, asthma, coronary artery disease, kidney disease, liver disease, obesity, cancer | Prevalence of ICU admission and death. | Unadjusted RR calculated |
| | | | | | | | Association of risk factors with death. | |
| Rivera-Izquierdo M et al., 2020 (PLOS ONE) | Spain | Granada | 3-16-2020 to 4-10-2020 | Retrospective | 238 | Sex, age, smoking, HTN, DM, CVD, COPD, CKD, active neoplasia, medications | Prevalence of death. | Adjusted HR |
| | | | | | | | Association of risk factors with disease severity. | |
| Shabrawishi M et al., 2020 (Plos One) | Saudi Arabia | Mecca | 3-12-2020 to 4-8-2020 | Retrospective | 150 | Age, sex, HTN, DM, CVD, CKD, hypothyroidism, cancer, CVA, COPD, CLD | Association of risk factors with disease severity and death. | Unadjusted RR calculated |
| Shahriarirad R et al., 2020 (BMC Infectious Diseases) | Iran | Fars Province | 2-20-2020 to 3-20-2020 | Multicenter Retrospective | 113 | Age, sex, HTN, DM, CVD, COPD, CKD, malignancy, other immunosuppressive diseases | Prevalence of death. | Unadjusted RR calculated |
| | | | | | | | Association of risk factors with disease severity. | |
| Shekhar R et al., 2020 (Infectious Diseases) | USA | New Mexico | 1-19-2020 to 4-24-2020 | Cohort | 50 | Age, sex, HTN, DM, COPD, alcoholic cirrhosis, alcohol use, obesity | Association of risk factors with disease severity. | Unadjusted RR calculated |
| Shi Y et al., 2020 (Crit Care) | China | Zhejiang province | Not specified to 02-11-2020 | Retrospective | 487 | Age, sex, smoking, HTN, DM, CKD, CVD, CLD, cancer | Prevalence of and association of risk factors with severe disease | Unadjusted RR calculated |
| Suleyman G et al., 2020 (JAMA Network) | USA | Metropolitan Detroit | 3-9-2020 to 3-27-2020 | Retrospective | 463 | Age, sex, ethnicity, COPD, obstructive sleep apnea, DM, HTN, CVD, CKD, cancer, rheumatologic disease, organ transplant, obesity, smoking | Association of risk factors with disease severity. | Adjusted OR |
| Sun L et al., 2020 (Journal of Medical Virology) | China | Beijing | 1-20-2020 to 2-15-2020 | Retrospective | 55 | Age, sex, exposure, HTN, DM, CVD, Lung Disease, CKD, CLD | Prevalence of severe disease. Association of risk factors with severe disease. | Unadjusted RR calculated |
| Tambe M et al., 2020 (Indian J Public Health) | India | Pune | 3-31-2020 to 4-24-2020 | Cross-Sectional | 197 | Age, sex, HTN, DM, COPD, CVS, ALD, CKD | Association of risk factors with disease severity and death. | Unadjusted RR calculated |
| Tian S et al., 2020 (Journal of Infection) | China | Beijing | 1-20-2020 to 2-10-2020 | Retrospective | 262 | Age, sex, contact history, exposure to Wuhan. | Prevalence of death. Association of severe disease with risk factors. | Unadjusted RR calculated |
| Tomlins J et al., 2020 (Journal of Infection) | UK | Bristol | 3-10-2020 to 3-30-2020 | Retrospective | 95 | Age, sex, HTN, DM, COPD, CVD, cancer, renal disease, gastrointestinal disease, neurological disease | Prevalence of death. Association of risk factors with death. | Unadjusted RR calculated |
| Turcotte J.J et al., 2020 (PLOS ONE) | USA | Maryland | 3-1-2020 to 4-12-2020 | Retrospective | 117 | Age, BMI, sex, DM, obstructive sleep apnea, COPD, CVD, CKD, HTN, smoking, alcohol use, liver disease | Association of risk factors with disease severity and death. | Adjusted OR |

(*Continued*)

**Table 1.** (*Continued*)

| Author, year of publication (journal) | Country | Region | Study Period | Study Design | Size | Epidemiological Risk Factor | Outcome | Measures of Association |
|---|---|---|---|---|---|---|---|---|
| Wan S et al., 2020 (Journal of Medical Virology) | China | Northeast Chongqing | 1-23-2020 to 2-8-2020 | Retrospective | 135 | Age, sex, smoking, CKD, COPD, HTN, DM, CVD, Cancer, CLD, exposure, travel history | Prevalence of severe disease. Association of risk factors with severe disease. | Unadjusted RR calculated |
| Wang D et al., 2020 (JAMA) | China | Wuhan | 1-1-2020 to 1-28-2020 | Retrospective | 138 | Age, sex, Huanan Seafood Market Exposure, HTN, DM, CVD, COPD, Cancer, CKD, CLD, HIV | Prevalence of death and ICU admission. Association of risk factors with severe disease (ICU care) | Unadjusted RR calculated |
| Wang R et al., 2020 (Internal Journal of Infectious Diseases) | China | Fuyang | 1-20-2020 to 02-09-2020 | Retrospective | 125 | Age, sex, CVD, Cancer | Prevalence of critical disease. Association of age, sex, and smoking with critical disease. | Unadjusted RR calculated |
| Wang Z et al., 2020 (CID) | China | Wuhan | 1-16-2020 to 01-29-2020 | Retrospective | 69 | Age, sex, HTN, DM, CVD, COPD, Cancer, HBV, Asthma | Prevalence of death and severe disease (SpO2<90%). Association of risk factors with severe disease. | Unadjusted RR calculated |
| Wei Y et al., 2020 (BMC Infectious Diseases) | China | Hubei Province | 1-27-2020 to 3-22-2020 | Retrospective | 276 | Age, sex, smoking, obesity, HTN, COPD, CVD, DM, cerebrovascular disease, cancer | Association of risk factors with disease severity. | Unadjusted RR calculated |
| Wu C et al., 2020 (JAMA Intern Med) | China | Wuhan | 12-15-2019 to 01-26-2020 | Retrospective | 201 | Age, sex, HTN, DM, CVD, CKD, Chronic Lung Disease, Cancer, CLD, Sea Food Market Exposure. | Prevalence of ARDS, ICU admission and death. Association of risk factors with severe disease (ARDS) and death. | Unadjusted RR calculated |
| Yang X et al, 2020 (Lancet Respir Med) | China | Wuhan | 12-24-2019 to 1-26-2020 | Retrospective | 52 | Age, sex, exposure, COPD, diabetes, chronic cardiac disease, smoking, malnutrition | Association of risk factors with death. | Unadjusted RR calculated |
| Yao Q et al., 2020 (Pol Arch Intern) | China | Huanggang, Hubei | 1-30-2020 to 2-11-2020 | Retrospective | 108 | Age, sex, smoking, HTN, DM, CVD, CLD, cancer | Prevalence of severe disease and death. Association of risk factors with severe disease and death. | Unadjusted RR calculated |
| Young BE et al., 2020 (JAMA) | Singapore | Singapore | 1-23-2020 to 2-3-2020 | Retrospective | 18 | Age, sex | Prevalence of severe disease (receiving supplemental O2). Association of severe disease with age and sex. | Unadjusted RR calculated |
| Yu T et al., 2020 (Clinical Therapeutics) | China | Guangdong | January to February 2020 | Cross-sectional | 95 | Age, sex, current smoker | Prevalence of ARDS. Association of age, sex, and smoking with ARDS. | Unadjusted RR calculated |
| Yu X et al., 2020 (Transboundary and Emerging Diseases) | China | Shanghai | Up to 2-19-2020 | Retrospective | 333 | Age, sex, BMI, smoking, alcohol, exposure, HTN, DM, CVD | Prevalence of death and severe disease (Severe/critical pneumonia). Association of risk factors with severe disease. | Adjusted OR for age group, sex, CVD, DM, HTN. |

(*Continued*)

**Table 1.** (Continued)

| Author, year of publication (journal) | Country | Region | Study Period | Study Design | Size | Epidemiological Risk Factor | Outcome | Measures of Association |
|---|---|---|---|---|---|---|---|---|
| Zhan T et al., 2020 (J Int Med Res) | China | Wuhan | 1-12-2020 to 3-8-2020 | Retrospective | 405 | Age, sex, smoking, alcohol history, CVD, gastrointestinal disease, COPD, CKD, CLD | Association of risk factors with disease severity. | Unadjusted RR calculated |
| Zhang G et al., 2020 BMC Respiratory Research) | China | Wuhan | 1-16-2020 to 2-25-2020 | Retrospective | 95 | Age, sex | Prevalence of severe disease, composite end point, and death. Association with severe disease. | Unadjusted RR calculated |
| Zhang J et al., 2020 (Clin Microbiol Infect) | China | Wuhan | 1-11-2020 to 2-6-2020 | Retrospective | 663 | Age, sex, COPD, CVD, gastrointestinal disease, CKD, cancer | Prevalence of death. | Adjusted OR |
| | | | | | | | Association of risk factors with disease severity. | |
| Zhang JJ et al., 2020 (Allergy) | China | Wuhan | 1-16-2020 to 2-3-2020 | Retrospective | 140 | Age, sex, current smoker, past smoker, exposure history, HTN, DM, CVD, COPD, CKD, CLD | Prevalence of severe disease. Association of risk factors with severe disease (ICU admission). | Unadjusted RR calculated |
| Zhao X-Y et al., 2020 (BMC Inf Dis) | China | Hubei (Non-Wuhan) | 1-16-2020 to 2-10-2020 | Retrospective | 91 | Age, sex, DM, COPD, Cancer, Kidney disease | Prevalence of death. Association of risk factors with severe disease | Unadjusted RR calculated |
| Zheng S et al., 2020 (BMJ) | China | Zhejiang province | 1-19-2020 to 2-15-2020 | Retrospective | 96 | Age, sex, HTN, DM, CVD, lung disease, Liver disease, renal disease, malignancy, viral Load, immunocompromise | Prevalence of death and severe disease. | Unadjusted RR calculated |
| | | | | | | | Association of risk factors with severe disease. | |
| Zheng Y et al., 2020 (Pharmacological Research) | China | Shiyan, Hubei | 1-16-2020 to 2-4-2020 | Retrospective | 73 | Age, sex, exposure, smoking history, DM, CVD | Prevalence of severe (severe/ critical) disease. Association of smoking and diabetes with severe disease. | Unadjusted RR calculated |
| Zhou F et al., 2020 (The Lancet) | China | Wuhan | 12-29-2019 to 1-31-2020 | Retrospective | 191 | Age, sex, current smoking, exposure history, HTN, DM, CVD, COPD, cancer, CKD | Prevalence of severe disease (ICU admission) and death. Association of risk factors with death. | Adjusted OR for age and CVD. Unadjusted RR calculated for other variables. |

CVD, cardiovascular disease; CKD, chronic kidney disease; CLD, chronic liver disease; COPD, chronic obstructive pulmonary disease; HTN, hypertension; DM, diabetes mellitus; ICU, intensive care unit; BMI, body mass index; HIV, human immunodeficiency virus; AIDS, acquired immunodeficiency syndrome; RR, relative risk; HR, hazard ratio; OR, odds ratio.

prevalence of death, severe disease, and risk factors (S1 Table), and association of the risk factors with death (S2 Table) and severe disease (S3 Table) for the individual studies are presented in the supplemental tables.

## Predictors of death and severe disease (Tables 2 and 3)

**Age and sex.** Median age for people who died was 79 years [IQR: 77–80; $I^2$ = 89%; n = 28] and who had severe disease was 61 years [IQR: 59–63; $I^2$ = 48%; n = 26]. Eighty-five percent [95% CI: 80–89; I2 = 76%; n = 18] of the deaths were in people aged $\geq$ 60 years and 66% [95% CI: 64–69; n = 34] were in males. The CFR (95% CI) was 35% (28–43%) for age$\geq$60 years and 26% (21–32%) for males. Patients aged$\geq$60 years [summary relative risk (sRR): 3.61; 95% CI:

**Table 2. Pooled prevalence of death stratified by epidemiological risk factors in COVID-19 patients hospitalized between December 2019-August 2020.**

| Risk factor or Outcome | Overall prevalence of risk across studies | | Pooled Prevalence of Death (Case Fatality Risk) and Risk Factor | | | Summary Relative Risk of Death | | | |
|---|---|---|---|---|---|---|---|---|---|
| | No. of studies | Pooled prevalence of risk factor and death, % (95% CI) | No. of studies | *Case fatality risk (Prevalence of death in risk group), % (95% CI) | #Prevalence of risk factor in persons who died, % (95% CI) | No. of studies | Fixed Effects | Random Effects# | Heterogeneity |
| | | | | | | | Summary relative risk; 95% CI (Shore adjusted) | sRR; (95% CI) | $I^2$; $c^2$; p value |
| Death | 60 | 20 (18–23) | N/A | N/A | N/A | N/A | N/A | N/A | N/A |
| Age $\geq$ 60 years | 41 | 48 (44–53) | 18 | 35 (28–43) | 85 (80–89) | 24 | 3.61 (2.96–4.39) | 1.29 (1.03–1.62) | 77%; 99; p<0.01 |
| Male | 75 | 59 (57–60) | 31 | 26 (21–32) | 66 (64–69) | 36 | 1.31 (1.22–1.40) | 1.34 (1.24–1.45) | 18%; 43; p = 0.17 |
| Smoking history | 41 | 26 (22–31) | 11 | 27 (24–32) | 44 (38–50) | 13 | 1.28 (1.06–1.55) | 1.41 (1.12–1.78) | 68%; 37; p<0.01 |
| Current smoker | 21 | 10 (7–13) | 7 | 21 (14–29) | 13 (7–24) | 8 | 1.43 (91–2.26) | 1.53 (95–2.45) | 78%; 32; p<0.01 |
| COPD | 52 | 9 (8–11) | 20 | 51 (36–71) | 12 (7–19) | 22 | 1.70 (1.42–2.04) | 1.74 (1.43–2.13) | 66%; 61; p<0.01 |
| Hypertension | 64 | 50 (46–54) | 29 | 28 (23–36) | 66 (61–70) | 32 | 1.76 (1.58–1.96) | 1.88 (1.66–2.13) | 56%; 70; p<0.01 |
| Diabetes | 67 | 28 (25–31) | 29 | 24 (17–33) | 39 (35–44) | 33 | 1.50 (1.35–1.66) | 1.60 (1.42–1.79) | 58%; 77; p<0.01 |
| Cardiovascular disease | 65 | 17 (15–20) | 29 | 52 (46–60) | 37 (32–43) | 34 | 2.08 (1.81–2.39) | 2.25 (1.92–2.64) | 69%; 106; p<0.01 |
| Chronic kidney disease | 47 | 13 (11–16) | 18 | 48 (37–63) | 27 (21–34) | 23 | 2.52 (2.11–3.00) | 2.39 (1.91–2.99) | 72%; 79; p<0.01 |
| Chronic Liver Disease | 31 | 2(2–3) | 8 | 39(31–50) | 6 (4–8) | 9 | 2.65(1.88–3.75) | 1.99 (1.26–3.16) | 77%; 35; p<0.01 |

*Case fatality risk of represent total number of people that died in the specific risk group divided by total population in the risk group.

# Prevalence of risk group in dead represent total number of people having the risk group divided by total population that died.

2.96–4.39; $I^2$ = 77%; n = 24] and males [sRR: 1.34; 95% CI: 1.22–1.40; $I^2$ = 18%; n = 36] had higher risk of death. The risk of severe disease was similarly higher for age>60 [sRR: 1.57; 95% CI: 1.36–1.80; $I^2$ = 85%; n = 29] and males [sRR: 1.26; 95% CI: 1.18–1.35; $I^2$ = 38%; n = 47].

**Hypertension.** The prevalence of hypertension in the COVID-19 patients was 50% [95% CI: 46–54% $I^2$ = 98%; n = 64], with a CFR in hypertensive patients of 28% [95% CI: 23–36%; $I^2$ = 97%; n = 29] and a CSR of 44% [95% CI: 37–53%; $I^2$ = 95%; n = 39]. Of the COVID-19 patients that died, 66% [95% CI: 61–70%; $I^2$ = 83%; n = 29] had hypertension. Hypertensives had higher relative risk of death [sRR: 1.76; 95% CI: 1.58–1.96; $I^2$ = 56%; n = 32] and severe disease [sRR: 1.46; 95% CI: 1.28–1.65; $I^2$ = 77%; n = 40] compared to non-hypertensives (Fig 2A).

**Diabetes.** The prevalence of diabetes was 28% [95% CI: 25–31%; $I^2$ = 97%; n = 67] with a CFR of 24% [95% CI: 17–33%; $I^2$ = 98%; n = 29] and CSR of 43% [95% CI: 38–49%; $I^2$ = 99%; n = 30] in the diabetics. Of the COVID-19 patients that died, 33% [95% CI: 32–44%; $I^2$ = 83%; n = 29] were diabetics. Diabetics had higher relative risk of death [sRR: 1.50; 95% CI: 1.35–1.66; $I^2$ = 58%; n = 33] and severe disease [sRR: 1.48; 95% CI: 1.35–1.63; $I^2$ = 59%; n = 44] compared to non-diabetics (Fig 2B).

**Cardiovascular disease.** The pooled prevalence of CVD was 17% [95% CI: 15–20%; $I^2$ = 96%; n = 65] with a CFR of 52% [95% CI: 46–60%; $I^2$ = 81%; n = 29] and CSR of 56% [95% CI: 48–65%; $I^2$ = 91%; n = 37] among cardiac patients. Of the patients that died, 37% [95% CI: 32–

**Table 3. Pooled prevalence of severe disease stratified by epidemiological risk factors in COVID-19 patients.**

| Risk group or outcome | Prevalence of Severe Disease (Case Severity Risk) and Risk Factors | | | Summary Relative Risk of Severe Disease | | | |
|---|---|---|---|---|---|---|---|
| | No. of studies | Prevalence of severe disease and case severity risk*, % (95% CI) | *Prevalence of risk factor in people with severe disease, % (95% CI) | No. of studies | Fixed Effects | Random Effects# | Heterogeneity |
| | | | | | sRR; 95% CI (Shore adjusted) | sRR; (95% CI) | $I^2$; $c^2$; p value |
| Severe disease | 25 | 20 (16–25) | N/A | N/A | N/A | N/A | N/A |
| Age ≥ 60 years | 26 | 48 (39–59) | 56 (52–61) | 29 | 1.57 (1.36–1.80) | 1.76 (1.50–2.07) | 85%; 184; p<0.01 |
| Male | 45 | 40 (34–47) | 63 (61–66) | 47 | 1.26 (1.18–1.35) | 1.33 (1.22–1.44) | 38%; 75; p<0.01 |
| Smoking history | 27 | 39 (34–46) | 26 (21–32) | 27 | 1.29 (1.18–1.42) | 1.32 (1.18–1.47) | 33%; 39; p = 0.05 |
| Current smoker | 13 | 38 (28–53) | 13 (9–20) | 15 | 1.52 (1.21–1.91) | 1.25 (94–1.66) | 75%;56; p<0.01 |
| COPD | 24 | 43 (35–52) | 14 (12–17) | 29 | 1.71 (1.49–1.97) | 1.83 (1.54–2.18) | 84%;179; p<0.01 |
| Hypertension | 39 | 44 (37–53) | 55 (50–61) | 40 | 1.46 (1.28,1.65) | 1.54 (1.33,1.78) | 77%;168; p<0.01 |
| Diabetes | 43 | 43 (38–49) | 33 (30–38) | 44 | 1.48 (1.35–1.63) | 1.64 (1.47–1.82) | 59%;104; p<0.01 |
| Cardiovascular disease | 37 | 56 (48–65) | 28 (24–33) | 38 | 1.54 (1.39–1.72) | 1.74 (1.52–1.98) | 77%;164; p<0.01 |
| Chronic kidney disease | 22 | 36 (33–40) | 26 (19–37) | 27 | 1.56 (1.31–1.86) | 1.42 (1.15–1.76) | 85%; 176; p<0.01 |
| Chronic Liver Disease | 12 | 43(32–57) | 5 (3–7) | 15 | 1.63 (1.23–2.15) | 1.66 (1.16–2.36) | 82%; 76; p<0.01 |

*Case severity risk represent total number of people developing severe disease in the specific risk group divided by total population in that risk group.

# Prevalence of risk factor in severe disease represent total number of people with the risk factor divided by total population with severe disease.

43%; $I^2$ = 83%; n = 29] had CVD. Patients with CVD had higher relative risk of death [sRR: 2.08; 95% CI: 1.81–2.39; $I^2$ = 69%; n = 34] and severe disease [sRR: 1.54; 95% CI: 1.39–1.72; $I^2$ = 77%; n = 38] compared to patients without CVD (Fig 2C).

**Smoking and COPD.** The prevalence of any history of smoking in the patients was 26% [95% CI: 22–31%; $I^2$ = 98%; n = 41]. For patients with smoking history, the CFR was 27% [95% CI: 24–32%; $I^2$ = 61%; n = 11] and CSR was 39% [95% CI: 34–46; $I^2$ = 78%; n = 27]. Compared to never smokers, patients with smoking history had higher relative risk of death [sRR: 1.28; 95% CI: 1.06–1.55; $I^2$ = 68%; n = 13] and severe COVID-19 disease [sRR: 1.29; 95% CI: 1.18–1.42; $I^2$ = 33%; n = 27] (Fig 3A). The prevalence of COPD was 9% [95% CI: 8–11%; $I^2$ = 94%; n = 52]. Patients with COPD had a CFR of 51% [95% CI: 43–59%; $I^2$ = 0%; n = 21]; CSR of 43% [95% CI: 35–52%; $I^2$ = 84%; n = 24]; a sRR of death of 1.70 [95% CI: 1.42–2.04; $I^2$ = 66%; n = 22] and of severe disease of 1.71 [95% CI: 1.49–1.97; $I^2$ = 84%; n = 29] (Fig 3B).

**Chronic kidney disease.** The prevalence of CKD was 13% [95% CI: 11–16%; $I^2$ = 96%; n = 47] with a CFR of 48% [95% CI: 37–63%; $I^2$ = 89%; n = 18] and CSR of 36% [95% CI: 33–40%; $I^2$ = 56%; n = 22] in CKD patients. CKD was present in 27% [95% CI: 21–34%; $I^2$ = 79%; n = 18] of all COVID-19 patients that died. CKD patients had higher relative risk of death [sRR: 2.52; 95% CI: 2.11–3.00; $I^2$ = 72%; n = 23] and severe disease [sRR: 1.56; 95% CI: 1.31–1.86; $I^2$ = 85%; n = 27] compared to non-CKD patients (Fig 3C).

**Chronic liver disease.** The prevalence of CLD was 2% [95% CI: 2–3%; $I^2$ = 72%; n = 31] with a CFR of 39% [95% CI: 31–50%; $I^2$ = 0%; n = 8] and CSR of 43% [95% CI: 32–57%; $I^2$ = 5%; n = 12] in CLD patients. CLD was present in 6% [95% CI: 4–8%; $I^2$ = 0%; n = 8] of the

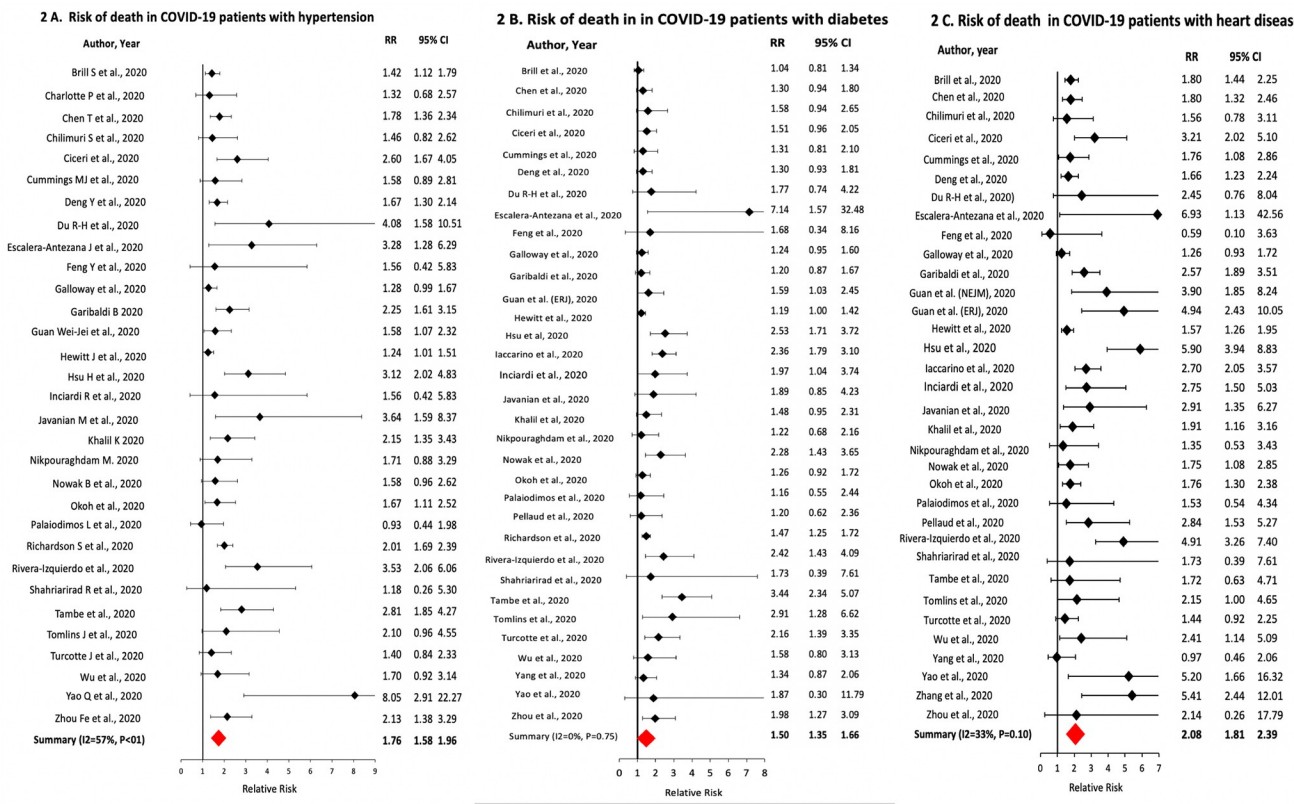

**Fig 2. Association of hypertension, diabetes and heart disease with death in COVID-19 patients.**

COVID-19 patients who died. Patients with CLD had higher relative risk of death [sRR: 2.65; 95% CI: 1.88–3.75; I² = 77%; n = 9] and severe disease [sRR: 1.63; 95% CI: 1.23–2.15; I² = 82%; n = 15] compared to non-CKD patients (Fig 3D).

## COVID-19 related organ system injury

To understand how pre-existing health conditions may be correlated with the risk of specific organ injury, we calculated the prevalence of acute injury to lung, heart and kidney for studies that reported prevalence of both the pre-existing condition(s) and corresponding organ injury (Fig 4A). Pooled across 12 studies [14, 25, 32, 45, 48, 49, 52, 54, 60, 62, 79], the prevalence of COPD at baseline was 6% [95% CI: 4–11%] and the proportion of patients developing ARDS during hospitalization was 48% [32–73%]. The pooled prevalence of baseline CVD (n = 13 studies) was 11% [95% CI: 9–15%] and that of acute cardiac injury (ACI) during hospitalization was 21% [95% CI: 15–28%] [6, 14, 25, 32, 35, 43, 48, 49, 54, 79, 84]. The prevalence of CKD (n = 12 studies) was 14% [95% CI: 8–26%] and that of acute kidney injury during hospitalization (AKI) was 27% [95% CI: 21–34%] [6, 14, 25, 32, 45, 48, 65, 79].

## Regional difference in prevalence of death and risk factors

Upon sub-group analysis, we noted significantly higher prevalence of death and risk factors among COVID-19 patients in the US and Europe than in China (Fig 4B). The prevalence of death was 23% [95% CI: 19–27%; I² = 97%; n = 29] in the US and Europe, and 11% [95% CI: 7–16%; I² = 94%; n = 24] in China. Prevalence of severe disease was 20% [95% CI: 16–25%;

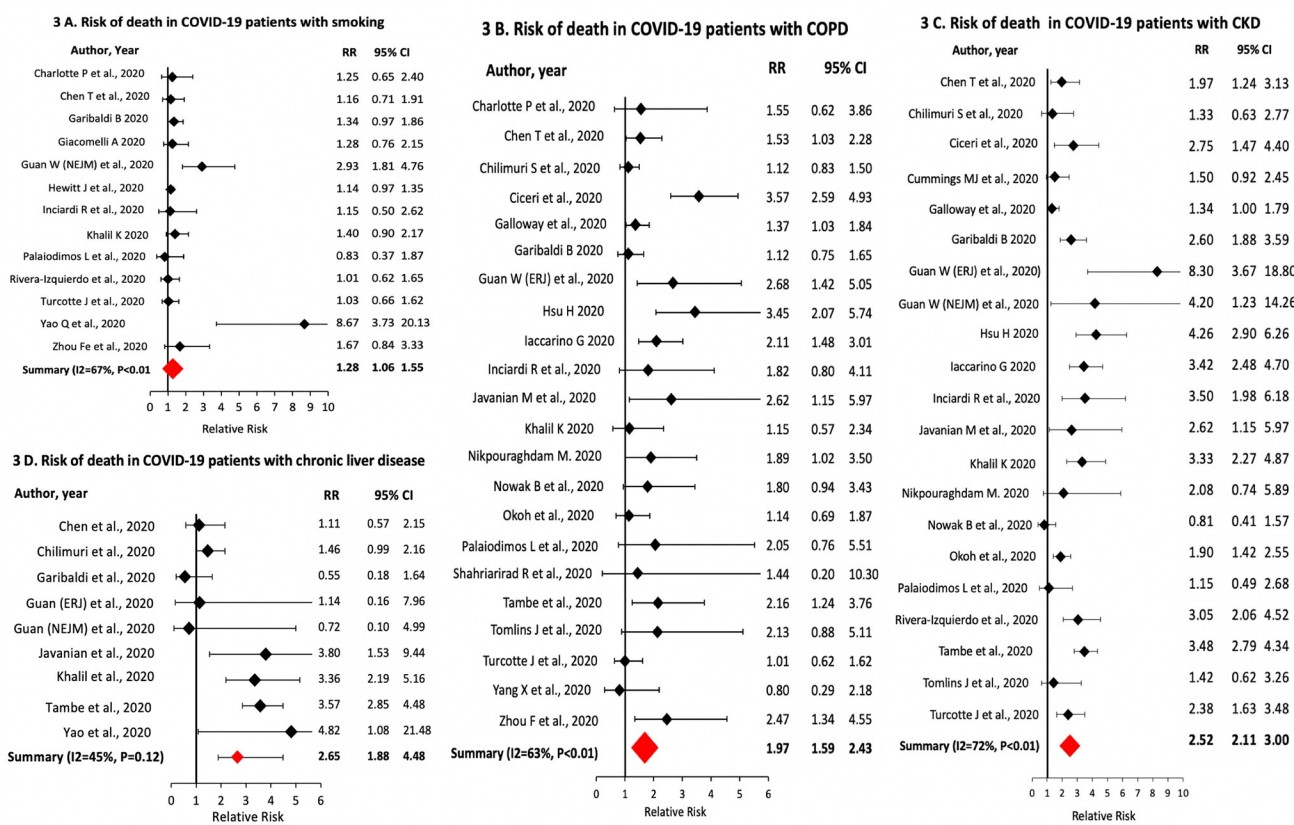

**Fig 3. Association of smoking, chronic obstructive pulmonary disease, chronic kidney disease and chronic liver disease with death in COVID-19 patients.**

$I^2 = 98\%$; n = 25] for US and Europe, and 39% [95% CI: 32–47%; $I^2 = 97\%$; n = 30] for China. Median age of patients was 65 years [IQR: 63–67 years; $I^2 = 0\%$; n = 24] for the US and Europe and 55 years [IQR: 52–58 years; $I^2 = 57\%$; n = 27] for China. Fifty-two percent [95% CI: 46–59%; $I^2 = 98\%$; n = 16] of the patients hospitalized were aged ≥60 years in the US and Europe as compared to 36% [95% CI: 30–43%; $I^2 = 96\%$; n = 22] for China. The prevalence of co-morbidities between US-Europe and China differed as follows: **1) US-Europe**: HTN = 55% [95% CI: 52–57%]; diabetes = 31% [95% CI: 29–34]; CVD = 18% [95% CI: 15–21%]; smoking history = 15% [95% CI: 11–21%]; COPD = 9% [95% CI: 6–13%] and **2) China**: HTN = 23% [95% CI: 20–26%]; diabetes = 12% [95% CI: 10–14%]; CVD = 16% [95% CI: 12–22%]; smoking history = 11% [95% CI: 9–13%]; CKD = 2.3% [95 CI: 1.6–3.4%] and COPD = 4% [95 CI: 3–5%].

## Comorbidities in COVID-19 patients and the general populations in the US and China

In order to gain some understanding of whether patients with comorbidities are at higher risk of COVID-19 infection or hospitalization, we compared the prevalence of comorbidities between COVID-19 patients hospitalized in the US and the prevalence of comorbidities in the general US population. We observed that the prevalence of hypertension (55%), diabetes (33%), CVD (17%), and smoking history (23%) were substantially higher in the COVID-19 patients than in the general US population (Fig 4C). For the Chinese population, the overall prevalence of hypertension (23%) and diabetes (12%) in the COVID-19 patients were similar

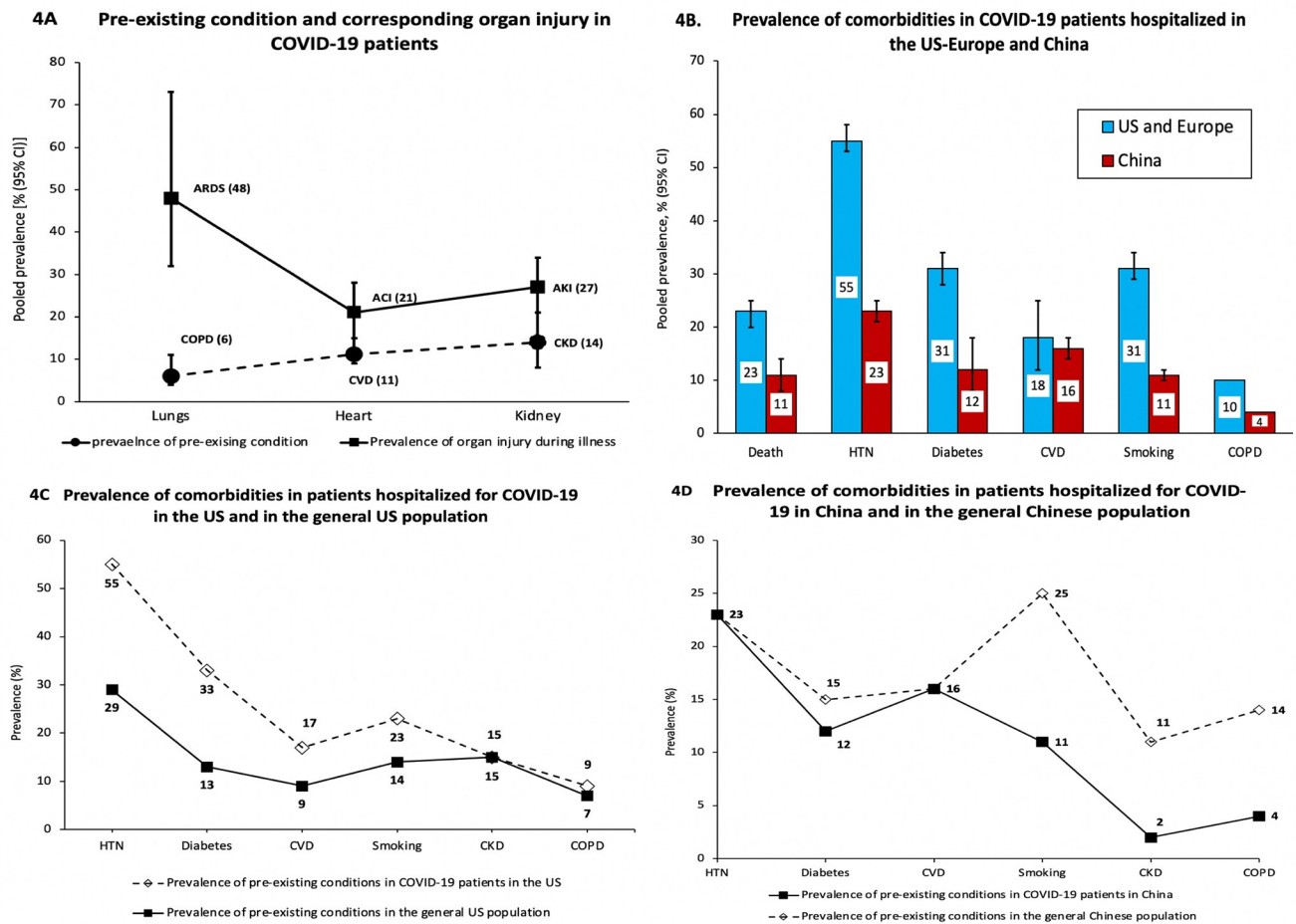

**Fig 4. Prevalence of acute organ injuries during hospital stay and regional difference in prevalence of death and comorbidities in patients hospitalized for COVID-19.** ARDS, acute respiratory distress syndrome; COPD, chronic obstructive pulmonary disease; ACI, acute cardiac injury; CVD, cardiovascular disease; AKI, acute kidney injury; CKD, chronic kidney disease; HTN, hypertension.

to that of the general Chinese population. However, the prevalence of smoking history (11%), COPD (4%), CKD (2%), and heart disease (16%) in the COVID-19 patients hospitalized in China were unexpectedly lower as compared to their corresponding prevalence in the general Chinese population (Fig 4D).

## Sensitivity analyses

The positive associations of age≥65 years, male sex, smoking history, COPD, hypertension and diabetes with the risk of death in the COVID-19 patients were relatively homogenous ($I^2$<70%). However, we carried out sensitivity analyses to assess the effects of outliers. For the risk of death for hypertension and smoking history, we removed the study by Yao et al. [86] which showed significantly higher risk compared to other studies; the results for both hypertension [sRR = 1.74; 95% CI: 1.58–1.94] and smoking [sRR:1.24; 95% CI: 1.08–1.42] remained significant. Guan et al. [13] had published a second study with additional patients and reported adjusted estimates for COPD, diabetes and hypertension. We used the adjusted risk estimates for the analyses. For the risk of death with other risk factors (CVD, CKD, and CLD) for Guan et al. [45], we conducted sensitivity analyses by using the counts only from the original study.

The results [sRR (95% CI)] were similar as: CVD = 2.06 [95 CI: 1.80–2.36], CKD = 2.48 [95% CI: 2.09–2.94] and CLD = 2.67 [95% CI: 1.85–3.85].

## Small study effects and quality assessment

We observed asymmetry in the funnel plot for studies that reported prevalence of death in COVID-19 patients (Egger's test p = 0.001) (S1 Fig). On further analysis, the plot remained asymmetrical when restricted to studies from China (Egger's p = 0.003) but was symmetrical for studies from US-Europe (Egger's p = 0.160). We observed symmetrical funnel plots with no bias for pooled prevalence severe disease (Egger's p = 0.128). On average, prospective or retrospective studies scored a score of 6 out of 9 and cross-sectional studies scored 6 out of 10. Many studies did not get a full score because they did not adjust for confounders (age, sex, or other risk factors) or patients remained hospitalized even after the follow-up ended, suggesting inadequate follow-up period (S4 Table).

## Discussion

We carried out a comprehensive systematic review and meta-analysis of 77 studies that included 38906 hospitalized patients to investigate the prevalence and risk factors for death and severe disease in COVID-19 patients. We calculated an overall prevalence of death of 20% and severe disease of 28%. Nearly 50% of the patients admitted to hospitals due to COVID-19 were ≥60 years of age and 59% were males. We observed high prevalence of hypertension and diabetes of 50% and 28%, respectively, for the patients. The risk factors were more prevalent in patients who died and were distributed as: age ≥60 years: 85%; males: 66%; hypertension: 66%; diabetes: 39%; heart disease: 37%; CKD: 27%; smoking history: 44%; COPD: 12%, and CLD: 9%. In comparison with the overall prevalence of death of 20% for all COVID-19 hospitalized patients, the CFR was higher for male patients (26%) and for patients having the following risk factors: age≥60 years (35%), heart disease (52%), COPD (51%), CKD (48%), CLD (39%), hypertension (28%), diabetes (24%), and smoking history (27%). The elevation in the risk of death was statistically significant for age ≥60 (sRR = 3.6; 95% CI: 3.0–4.4), male sex 1.3 (95% CI: 1.2–1.4), smoking history (sRR = 1.3; 95% CI: 1.1–1.6), COPD (sRR = 1.7; 95% CI: 1.4–2.0), heart disease (sRR = 2.1; 95% CI: 1.8–2.4), CKD (sRR = 2.5; 95% CI: 2.1–3.0), hypertension (sRR = 1.8; 95% CI: 1.7–2.1), and diabetes (sRR = 1.5; 95% CI: 1.4–1.7). All of the risk factors we analyzed were positively associated with progression to severe disease as well. The results suggest that older age, male sex and the co-morbidities increase the risk of progression to severe disease and death in COVID-19 patients.

We observed significant difference in the prevalence of death between US-Europe (23%) and China (11%). This lower risk of death from COVID-19 for the hospitalized patients in China may be explained by the lower median age as well as lower prevalence of co-morbidities for COVID-19 patients in China. However, this >200% lower prevalence of death in China is incommensurate with our finding of a higher prevalence of severe disease observed for patients in China (39%) as compared to patients in the US-Europe (20%). Notably, we observed asymmetry in the funnel plot and a statistically significant tests for publication bias or small study effects for the prevalence of death for studies from China that could suggest selective outcome reporting. As such, while the lower median age and prevalence of co-morbidities for COVID-19 patients in China may explain the lower prevalence of death, it is also possible that a selective under-reporting of death had occurred for studies from China. The death toll in China was initially under-reported and later updated on April 17, 2020 [95].

Whether or not cigarette smoking has been associated with SARS-CoV-2 acquisition or progression to severe disease has been strongly debated with studies showing both positive,

null, and inverse association between smoking and COVID-19 [10, 11, 96–98]. We found that patients with any history of smoking have both a higher risk of death (RR: 1.28; 95% CI: 1.06–1.55) and severe disease (1.29; 95% CI: 1.18–1.42). The case fatality risk for those with smoking history (27%) was also higher than the overall CFR of 20%. Whereas a higher COVID-19 mortality and morbidity among smokers may be due its causal association with COPD and CVD, Cai et al. [99] has also observed upregulation of pulmonary Angiotensin Converting Enzyme 2 (ACE2) gene expression and hence, pulmonary ACE2 receptors in smokers suggesting a direct effect of smoking on COVID-19 susceptibility and disease progression. ACE2 receptors are used by SARS-CoV-2 to translocate intracellularly [15, 100–104].

Our results of higher risk of death and severe disease associated with hypertension, diabetes and CVD in COVID-19 patients concurred with most studies conducted to date including studies that specifically investigated these associations [14, 65, 105, 106]. However, it is unclear if cardiovascular risk factors including smoking, hypertension, diabetes, heart disease and CKD increases the susceptibility toward SARS-CoV-2 infection in the population [15, 100, 101, 107]. On one hand, angiotensin-converting enzyme 2 (ACE2)–by blocking the renin angiotensin aldosterone system (RAAS) and decreasing or countering the vasoconstrictive, proinflammatory and profibrotic properties of angiotensin-II through catalysis of angiotensin-II to angiotensin-(1–7)–have been shown to exert cardiovascular protective effect and prevent acute lung injury from SARS-CoV-2 [15, 100, 101]. However, on the other hand, a possible greater expression of ACE2, the functional receptor mediating cellular entry of SARS-CoV-2 in humans, in patients with cardiovascular disease and other comorbidities can lead to increased susceptibility towards infection with SARS-CoV-2 [108, 109]. In this context, it would be reasonable to posit that a substantially higher prevalence of cardiovascular comorbidities in the hospitalized patients compared to the prevalence in the general population may suggest elevated risk of acquisition of SARS-CoV-2 for patients with cardiovascular risk factors. To this end, we found that the prevalence of smoking history (23%), hypertension (55%), diabetes (33%) and heart disease (17%) in the hospitalized COVID-19 patients in the US were substantially higher than the corresponding prevalence of smoking (14%) [110], hypertension (29%) [111], diabetes (13%) [112] and heart disease (9%) [113] in the general US population that could suggest an association between these comorbidities and risk of SARS-CoV-2 infection or disease progression. However, we note that if the prevalence of these comorbidities in the asymptomatic individuals with COVID-19 in the general population is similar to that of their prevalence in the non-COVID-19 general population, then this difference–the higher prevalence of comorbidities in the hospitalized patients compared to the general population–could simply imply a higher risk of symptomatic infection or hospitalization for individuals having SARS-CoV-2 infection. The prevalence of other risk factors i.e. COPD (9%) and CKD (15%) in the COVID-19 patients in the US was similar to the overall prevalence of COPD (7%) [114] and CKD (15%) [115] in the country. Generally, we noted a lower prevalence of comorbidities for patients in China. The prevalence of hypertension (23%) and diabetes (12%) in the hospitalized patients in China, which were lower than that of the US, approximate the respective prevalence of hypertension (23%) [116] and diabetes (15%) [117] in the general population of China. A previous meta-analysis also noted this observation [19]. Surprisingly, the prevalence of smoking (11%) in the COVID-19 patients hospitalized in China are inexplicably lower than the corresponding prevalence of smoking (23%) among COVID-19 patients in the US despite a higher prevalence of smoking (47% in Chinese males) [118] in the general Chinese population is significantly higher than that of the US. The prevalence of CVD (16%), COPD (4%) and CKD (2%) among COVID-19 patients in China are substantially lower than the corresponding prevalence of CVD (21%) [119], COPD (14%) [120], and CKD (11%) [121] in the general Chinese population. Given these discrepancies, we are unsure whether the lower

prevalence of comorbidities noted for the COVID-19 patients in China are representative of the true prevalence. There was a great sense of urgency and a race to publish data in the early phase of the outbreak. As such, there exists the possibility of substantial under-recording of data on covariables. Had there been under-reporting, the implication would be a higher true prevalence estimate. We do not see reason for any systematic difference in reporting of risk factors based on outcome, or vice-versa, and hence, our summary relative risk estimates for association of risk factors with death or severe disease should not have been affected.

We assessed if patients with specific co-morbidities at baseline had higher risk of specific organ injury from SARS-CoV-2 during hospitalization. While the available data did not allow direct assessment of this relation, we compared the prevalence of comorbidities with the prevalence of corresponding organ system injury for studies that reported both baseline comorbidity and corresponding organ injury. We observed that the risk of acute lung injury/ARDS (48%), ACI (21%), and AKI (27%) were substantially higher than the baseline prevalence of COPD (6%), heart disease (11%) and CKD (14%), respectively. The higher prevalence of acute organ injury than the prevalence of baseline comorbidity simply indicates that ARDS, ACI and AKI were also occurring in patients who did not have a corresponding comorbidity at baseline in addition to people having the comorbidities.

Most studies reported only frequencies of risk factors and did not present adjusted measures for disease severity or death. Given this limitation, the risk ratio we calculated from the frequencies are largely unadjusted estimates. Future studies could additionally present, at the least, age- and sex-adjusted measures for association of risk of comorbidities with death or severe disease. Many studies reported odds ratio for the measure of association between pre-existing conditions and risk of severe disease or death. Odds ratio poorly approximates risk ratio when the disease prevalence is high at baseline. For example, Zhou et al. [14] calculated an odds ratio of 5.4 (95% CI: 0.96–30.4) for risk of death from COPD in COVID-19 patients whereas the risk ratio we calculated from the frequencies presented is RR = 2.47 (95% CI: 1.34–4.55). Prevalence of severe disease or death in COVID-19 patients was high in several studies. Similarly, several meta-analyses calculated odds ratios instead of risk ratios to summarize the risk of disease severity or death in association with risk factors such as smoking, diabetes, hypertension and cardiovascular disease [10, 11, 18], often to be interpreted by media and even by researchers as a measure of relative risk. Lack of rigor in research design, analysis and interpretation could generate inconsistent and ungeneralizable results across studies leading to controversy and confusion around serious public health issues such as that existing for association (or not) of smoking with COVID-19 disease acquisition, severity or death. As publications evolve at a pace that could be overwhelming for researchers and practitioners, we attempted to present a meaningful summary and inference for association of risk factors with death or severe disease from literatures published globally. Additionally, we provide an epidemiological framework for the risk of infection by SARS-CoV-2 based on presence of cardiovascular risk factors. This analysis can inform public health measures for COVID-19 screening and prevention, risk stratification and management of patients in clinical practice, analysis and presentation strategies for research data and inspire etiological investigations.

## Conclusion

Epidemiological risk factors for progression of COVID-19 to severe disease and death and for acquisition of SARS-CoV-2, the causal agent for COVID-19, based on presence of pre-existing conditions have been insufficiently understood. Meta-analysis of 77 studies including 39023 COVID-19 patients hospitalized globally revealed case fatality risk of 52% for those having heart disease, 51% for COPD, 48% for CKD, 39% for CLD, 28% for hypertension, 27% for smoking

history, 24% for diabetes, 35% for age≥60 years, and 26% for males. Of all the patients who died, an overwhelming majority (85%) were in people aged≥60 years. Also, of the people who died, 66% were males, 66% had hypertension, 44% had history of smoking, 39% had diabetes, 37% had CVD, 27% had CKD, and 6% had CLD. All of the above risk factors were significantly associated with death and severe disease in the patients hospitalized for COVID-19. The prevalence of ARDS was 48%, ACI 21%, and AKI 28% in the hospitalized patients. A higher prevalence of hypertension, diabetes, smoking and heart disease in the COVID-19 inpatients as compared to that of the general population could imply a higher risk of SARS-CoV-2 infection or disease progression for patients having these risk factors. These findings could inform public health strategies for targeted screening and appropriate control of modifiable risk factors such as smoking, hypertension, and diabetes to reduce morbidity and mortality. Finally, based on the published literature, there were vast differences in the prevalence of death and risk factors for the populations in China and in US-Europe that should be further investigated.

## Supporting information

**S1 Table. Prevalence of death, severe disease and risk factors in COVID-19 patients (December 2019-August 2020).**
(DOCX)

**S2 Table. Prevalence of death stratified by risk factors in COVID-19 patients (December 2019-August 2020).**
(DOCX)

**S3 Table. Prevalence of severe disease stratified by risk factors in COVID-19 patients (Dec 2019-August 2020).**
(DOCX)

**S4 Table. Newcastle-Ottawa quality assessment (modified) for studies[#].** [#]Award of Points: Selection: points were awarded based on representativeness of the exposed group and unexposed group (2 points), ascertainment of exposures (1 point), and demonstration that outcome of interest was not present at the start of the study (1 point). Comparability (2 points): points were awarded based on whether the analyses were adjusted for age, sex, and other risk factors (2 points for adjustment to age and sex). Outcome (3points): points were awarded based on ascertainment of outcome through record linkage or independent blind assessment (1 points); duration of follow-up (1 point) (hospitalization till discharge); and adequacy of follow up for study population (complete follow up for the patients (vs whether patients were currently under treatment at the time of study report) (1 point), or if the patients currently under admission are excluded from outcome assessment (1 point).
(DOCX)

**S1 Fig. Publication bias or small study effects for prevalence of death and severe disease.**
(TIF)

**S1 Checklist. PRISMA 2009 checklist.**
(DOC)

## Author Contributions

**Conceptualization:** Kunchok Dorjee.

**Data curation:** Kunchok Dorjee, Hyunju Kim, Elizabeth Bonomo, Rinchen Dolma.

**Formal analysis:** Kunchok Dorjee, Hyunju Kim, Elizabeth Bonomo.

**Investigation:** Kunchok Dorjee, Hyunju Kim, Elizabeth Bonomo, Rinchen Dolma.

**Methodology:** Kunchok Dorjee, Hyunju Kim, Elizabeth Bonomo, Rinchen Dolma.

**Project administration:** Elizabeth Bonomo.

**Software:** Kunchok Dorjee, Elizabeth Bonomo, Rinchen Dolma.

**Validation:** Hyunju Kim, Elizabeth Bonomo, Rinchen Dolma.

**Visualization:** Elizabeth Bonomo.

**Writing – original draft:** Kunchok Dorjee.

**Writing – review & editing:** Kunchok Dorjee, Hyunju Kim, Elizabeth Bonomo, Rinchen Dolma.

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
