## [Decision Letter · Decision Letter 0]

21 Aug 2020

PONE-D-20-19976

Epidemiological Risk Factors Associated with Death and Severe Disease in Patients Suffering From COVID-19: A Comprehensive Systematic Review and Meta-analysis

PLOS ONE

Dear Dr. Dorjee,

Thank you for submitting your manuscript to PLOS ONE. After careful consideration, we feel that it has merit but does not fully meet PLOS ONE’s publication criteria as it currently stands. Therefore, we invite you to submit a revised version of the manuscript that addresses the points raised during the review process.

We look forward to receiving your revised manuscript.

Kind regards,

Davide Bolignano, MD, PhD

Academic Editor

PLOS ONE

Journal Requirements:

3. As research on this topic is proceeding at a fast pace, we would recommend that the search is updated to allow an up-to-date view of the topic.

5. Thank you for stating the following at the end of your manuscript:

'Funding: Dr. Dorjee is supported by grants from private philanthropists; the Johns Hopkins Alliance for Healthier World (Grant # 80045453); National Institute of Allergy and Infectious Diseases of the National Institute of Health (Grant # K01AI148583); the United Nations STOP TB PARTNERSHIP TB REACH (Grant # 134126); and the Pittsfield Anti-tuberculosis Association (PATA).'

'The author(s) received no specific funding for this work.'

6. Please include a copy of Table 3 which you refer to in your text.

Reviewers' comments:

Reviewer's Responses to Questions

**Comments to the Author**

1. Is the manuscript technically sound, and do the data support the conclusions?

Reviewer #1: Yes

Reviewer #2: Yes

2. Has the statistical analysis been performed appropriately and rigorously? 

Reviewer #1: Yes

Reviewer #2: Yes

3. Have the authors made all data underlying the findings in their manuscript fully available?

Reviewer #1: Yes

Reviewer #2: Yes

4. Is the manuscript presented in an intelligible fashion and written in standard English?

Reviewer #1: Yes

Reviewer #2: Yes

5. Review Comments to the Author

Reviewer #1: Authors showed results from systematic literature review regarding SARS-C0-V2 and outcomes.

They used PRISMa methodology; the article is presented in an intelligible fashion and is written in standard English.

In general the manuscript has typical content as for review with accepted style. My points are listed below. Conclusions and limitations are acceptable.

-some typhos exist needing correction: e.g. once angiotensin begins with capital letter other time with small letter

- citations are missing journel pages, for the instance ref.6, 7.

Reviewer #2: In this manuscript the Authors present the results of a fixed- effects meta-analysis designed to investigate the association between clinical/epidemiological risk factors and progression to death in patients hospitalized due to COVID-19. A total of 44 studies were included in the analysis, comprising 20594 hospitalized patients mainly from China and US and small cohorts from Italy, UK, Iran and Singapore. Despite the heterogeneity of the studies and the lack of adjusted measures for disease severity or death, the rationale behind the study is interesting and overall the paper is original and well written. The comparison between the prevalence of comorbidities/risk factors in hospitalized COVID-19 patients and in the general population is well summarized.

We feel that the Authors should blunt the link with cardiovascular diseases as the statement “…suggesting either a tendency for SARS-CoV-2 to more effectively establish infection in cardiovascular patients or to cause more severe disease among them prompting hospitalization” (page 20) does not seem quite convincing. This also suggests that the Discussion should better elaborate on the proportion of asymptomatic individuals and the interactions of comorbidities.

6. PLOS authors have the option to publish the peer review history of their article (what does this mean?). If published, this will include your full peer review and any attached files.

Reviewer #1: **Yes: **Mariusz Kusztal

Reviewer #2: No

---

## [Author Response · Author response to Decision Letter 0]

13 Oct 2020

A letter with detailed response to the feedback from editors and revised are being submitted herewith this submission.

---

## [Decision Letter · Decision Letter 1]

30 Oct 2020

PONE-D-20-19976R1

Prevalence and Predictors of Death and Severe Disease in Patients Hospitalized Due To COVID-19: A Comprehensive Systematic Review and Meta-analysis of 77 Studies and 38,000 patients.

PLOS ONE

Dear Dr. Dorjee,

Thank you for submitting your manuscript to PLOS ONE. After careful consideration, we feel that it has merit but does not fully meet PLOS ONE’s publication criteria as it currently stands. Therefore, we invite you to submit a revised version of the manuscript that addresses the points raised during the review process.

We look forward to receiving your revised manuscript.

Kind regards,

Davide Bolignano, MD, PhD

Academic Editor

PLOS ONE

Reviewers' comments:

Reviewer's Responses to Questions

**Comments to the Author**

1. If the authors have adequately addressed your comments raised in a previous round of review and you feel that this manuscript is now acceptable for publication, you may indicate that here to bypass the “Comments to the Author” section, enter your conflict of interest statement in the “Confidential to Editor” section, and submit your "Accept" recommendation.

Reviewer #1: All comments have been addressed

Reviewer #2: (No Response)

2. Is the manuscript technically sound, and do the data support the conclusions?

Reviewer #1: Yes

Reviewer #2: Yes

3. Has the statistical analysis been performed appropriately and rigorously? 

Reviewer #1: Yes

Reviewer #2: Yes

4. Have the authors made all data underlying the findings in their manuscript fully available?

Reviewer #1: Yes

Reviewer #2: Yes

5. Is the manuscript presented in an intelligible fashion and written in standard English?

Reviewer #1: Yes

Reviewer #2: Yes

6. Review Comments to the Author

Reviewer #1: The authors responded to all queries sufficiently. In recent version it is publishable in PLOS one.

Reviewer #2: In this Manuscript the Authors present a Systematic Review and Meta-analysis of prevalence and predictors of death and severe disease in patients hospitalized due to COVID-19. They performed a fixed-effects meta-analysis using Shore’s adjusted confidence intervals to address heterogeneity. The paper is technically sound, well presented and adequately discussed.

Occasional formatting errors should be fixed.

7. PLOS authors have the option to publish the peer review history of their article (what does this mean?). If published, this will include your full peer review and any attached files.

Reviewer #1: No

Reviewer #2: No

---

## [Author Response · Author response to Decision Letter 1]

3 Nov 2020

Only formatting issues have been pointed out. We have addressed the formatting issues.

---

## [Decision Letter · Decision Letter 2]

18 Nov 2020

Prevalence and Predictors of Death and Severe Disease in Patients Hospitalized Due To COVID-19: A Comprehensive Systematic Review and Meta-analysis of 77 Studies and 38,000 patients.

PONE-D-20-19976R2

Dear Dr. Dorjee,

We’re pleased to inform you that your manuscript has been judged scientifically suitable for publication and will be formally accepted for publication once it meets all outstanding technical requirements.

Kind regards,

Davide Bolignano, MD, PhD

Academic Editor

PLOS ONE

Additional Editor Comments (optional):

Reviewers' comments:

Reviewer's Responses to Questions

**Comments to the Author**

1. If the authors have adequately addressed your comments raised in a previous round of review and you feel that this manuscript is now acceptable for publication, you may indicate that here to bypass the “Comments to the Author” section, enter your conflict of interest statement in the “Confidential to Editor” section, and submit your "Accept" recommendation.

Reviewer #1: All comments have been addressed

Reviewer #2: (No Response)

2. Is the manuscript technically sound, and do the data support the conclusions?

Reviewer #1: Yes

Reviewer #2: (No Response)

3. Has the statistical analysis been performed appropriately and rigorously? 

Reviewer #1: Yes

Reviewer #2: (No Response)

4. Have the authors made all data underlying the findings in their manuscript fully available?

Reviewer #1: Yes

Reviewer #2: (No Response)

5. Is the manuscript presented in an intelligible fashion and written in standard English?

Reviewer #1: Yes

Reviewer #2: (No Response)

6. Review Comments to the Author

Reviewer #1: In my opinion all suggestions regarding manuscript improvement has been adressed. The paper in the current form can be published

Reviewer #2: (No Response)

7. PLOS authors have the option to publish the peer review history of their article (what does this mean?). If published, this will include your full peer review and any attached files.

Reviewer #1: No

Reviewer #2: No

---

## [Editor Report · Acceptance letter]

24 Nov 2020

PONE-D-20-19976R2 

Prevalence and Predictors of Death and Severe Disease in Patients Hospitalized Due To COVID-19: A Comprehensive Systematic Review and Meta-analysis of 77 Studies and 38,000 patients. 

Dear Dr. Dorjee:

I'm pleased to inform you that your manuscript has been deemed suitable for publication in PLOS ONE. Congratulations! Your manuscript is now with our production department. 

Kind regards, 

on behalf of

Dr. Davide Bolignano 

Academic Editor

PLOS ONE